# Exosomes from the Uterine Cavity Mediate Immune Dysregulation via Inhibiting the JNK Signal Pathway in Endometriosis

**DOI:** 10.3390/biomedicines10123110

**Published:** 2022-12-02

**Authors:** Ying Jiang, Xiaoshan Chai, Shengnan Chen, Zhaoying Chen, Hao Tian, Min Liu, Xianqing Wu

**Affiliations:** Department of Obstetrics and Gynecology, The Second Xiangya Hospital of Central South University, Changsha 430011, China

**Keywords:** exosomes, uterine aspirate fluid, microRNAs, macrophages polarization, immune dysregulation, endometriosis

## Abstract

Endometriosis is a chronic inflammatory disease with an uncertain pathogenesis. Peritoneal immune dysregulation plays an important role in the pathogenesis of endometriosis. Exosomes are messengers of intercellular communication. This study mainly investigated the role of exosomes from the uterine cavity in endometriosis. Exosomes of the uterine aspirate fluid were isolated and cocultured with macrophages for 48 h. Flow cytometry was used to detect macrophage polarization. A Human MAPK Phosphorylation Antibody Array and Western blot were used to detect the phosphorylation of the MAPK pathway. A microRNA sequencing analysis was used to detect differentially expressed miRNAs. Our research found that exosomes of the uterine aspirate fluid from endometriosis could reduce the proportion of CD80^+^ macrophages. Additionally, it could inhibit the expression of P-JNK in macrophages. However, the JNK activator anisomycin could increase the proportion of CD80^+^ macrophages. In addition, exosomes of the uterine aspirate fluid from endometriosis could promote the migration and invasion of endometrial stromal cells by acting on macrophages. The expression of miR-210-3p was increased in both exosomes and the eutopic endometrium in patients with endometriosis through miRNA sequencing, which could also reduce the proportion of CD80^+^ macrophages. In summary, we propose that exosomes from the uterine cavity in patients with endometriosis may affect the phenotype of macrophages by inhibiting the JNK signaling pathway, thus mediating the formation of an immunological microenvironment conducive to the development of endometriosis.

## 1. Introduction

Endometriosis (EMs) is a chronic inflammatory disease that can lead to chronic pelvic pain and infertility [1]. It can be diagnosed via clinical diagnosis, imaging techniques, and surgical diagnosis. Endometriosis is often misdiagnosed due to a lack of early diagnostic methods [2]. It is of great significance to explore the pathogenesis of endometriosis.

Endometriosis has a contentious etiology which is often considered to be related to retrograde menstruation, coelomic metaplasia, and lymphatic and vascular metastasis [3]. Endometriosis is a benign disease, but its biological behavior of metastasis is similar to malignant tumors [4]. Before tumor metastasis, tumor cells may secrete exosomes to promote the formation of the premetastatic niche [5,6,7,8,9]. Endometrium contains stem cells [10]. In endometriosis, endometrial stem cells may reflux into the pelvic cavity via retrograde menstruation and find suitable “soil” for seeding [10]. Immunological dysfunction caused by peritoneal macrophages is closely related to endometriosis [11]. We speculated whether the eutopic endometrium could also secrete exosomes in advance to promote the formation of “soil” suitable for the growth of an ectopic lesion.

Exosomes are microscopic membrane vesicles with a diameter of 30–150 nm that are mainly composed of lipids, proteins, and nucleic acids [12]. Exosomes may be isolated from various bodily fluids such as blood, ascites, urine, and amniotic fluid [12]. Exosomes can mediate cell-to-cell communication through transporting functional proteins, RNA, and lipids, making them a research hotspot in recent years [13,14]. The biological function of exosomes is dependent on exosomal microRNAs (miRNAs). Exosomal miRNAs can participate in a range of pathophysiological processes and have the potential as diagnostic and therapeutic biomarkers [15]. Furthermore, studies have shown that miRNAs may be involved in the formation of a premetastatic niche [16,17,18].

Exosomes are messengers of intercellular communication. Campoy et al. successfully isolated exosomes from the fluid fraction of uterine aspirates, which may reflect the physiologic status of the uterine cavity [19]. The isolation of exosomes from the uterine aspirates provides a novel biomarker for gynecological diseases. Since exosomes may be involved in the formation of the premetastatic niche in tumor metastasis, it is worth exploring whether the eutopic endometrium may secrete exosomes to promote the establishment of an immune microenvironment conducive to the development of endometriosis. In our study, the primary purpose was to search for exosomal miRNAs with potential biological value in endometriosis and to investigate whether exosomes from the uterine cavity could participate in the development of endometriosis by mediating immune dysregulation.

## 2. Materials and Methods

### 2.1. Sample Collection

This study was approved by the Ethics Committee of The Second Xiangya Hospital of Central South University (#2020-065). All the patients included in this study signed informed consent forms. The EMs group (n = 22) included patients who underwent laparoscopic surgery for ovarian endometriosis at the Second Xiangya Hospital of Central South University between January 2020 and August 2021. The average age of the EMs group was 29.32 years. All the patients with endometriosis were at stage III/IV according to the revised American Society for Reproductive Medicine classification system. The control group (n = 25) included patients who underwent surgical treatment for other benign gynecological diseases except for endometriosis, such as simple ovarian cyst, uterine fibroids, and tubal ligation. The average age of the control group patients was 28.44 years. All included patients had a regular menstrual cycle and had not taken any hormone drugs or been placed on an intrauterine device for at least 6 months before surgery. Patients with malignant tumors, autoimmune diseases, hyperthyroidism, severe liver and kidney insufficiency, infectious diseases, or psychiatric diseases were excluded.

After the vulva, vagina, and cervix were routinely disinfected, a disposable uterine tissue suction catheter (Nuo De Medical, Nanchang, China) was used to aspirate approximately 1 mL of uterine aspirates. Then, the uterine aspirates were diluted in equal proportions of phosphate-buffered saline (PBS). Next, the samples were centrifuged at 4 °C, 2500× *g* for 20 min to remove endometrial tissues and cell debris. The endometrial tissues were stored at −80 °C for miRNA sequencing. The supernatant was filtered through a 0.22 μm filter (Millipore, Darmstadt, Germany) to obtain the uterine aspirate fluid. The uterine aspirate fluid was stored at −80 °C for exosome extraction.

### 2.2. Isolation of Exosomes

Exosomes were isolated using the exosome isolation kit (Echobiotech, Beijing, China). This kit is based on the principle of size exclusion chromatography (SEC) to separate exosomes. Yang et al. showed that SEC was more suitable for RNA sequencing in extracellular vesicles [20]. We separated the exosomes according to the instructions. First, the uterine aspirate fluid was centrifuged at 3000 rpm for 10 min. The precipitation was discarded and the supernatant was filtered through a 0.22 μm filter (Millipore, Darmstadt, Germany). Then, the sample was added to the top of the SEC column after rinsing the column with 20 mL of sterile PBS. After the sample was completely flowing into the SEC column, PBS was added for elution. After the sample was completely flowing into the SEC column, 1.5 mL of PBS was added to the elute. The collected fraction at this time contained no exosomes and was discarded. Next, 2.5 mL of PBS were added to the SEC column, and this fraction containing exosomes was collected. The collected fractions were centrifuged at 4000× *g* for 10 min with a 100 KDa Amicon Ultra Centrifugal Filter (Millipore, Darmstadt, Germany) to obtain exosomes.

### 2.3. Transmission Electron Microscopy (TEM)

A total of 10 μL of exosomes were dropped on a copper mesh. After incubation at room temperature for 10 min, the copper mesh was washed three times with ultrapure water. Then, 2% uranyl acetate was used to stain the exosomes for 1 min. The morphology of the exosomes was observed with a TEM (Hitachi H-7650, Tokyo, Japan) at 80 KV after the copper mesh was dried thoroughly.

### 2.4. Nanoparticle Tracking Analysis (NTA)

The concentration of exosomes was measured with a 405 nm laser by ZetaView PMX 110 (Particle Metrix, Munich, Germany). Then, the concentration of the exosomes was diluted from 1 × 10^7^ particles/mL to 1 × 10^9^ particles/mL with PBS to detect the size and quality of the exosomes.

### 2.5. Western Blot

In brief, the exosomes and collected cells were lysed with a RIPA lysis buffer (Beyotime, Shanghai, China) to extract the proteins. The total protein concentration was detected using the BCA Protein Assay kit (Beyotime, Shanghai, China). Then, the protein was separated using SDS-PAGE and was transferred to polyvinylidene fluoride (PVDF) membranes. The PVDF membrane was blocked with 5% skim milk powder at room temperature for 1 h. Next, the membrane was incubated with primary antibodies: anti-CD63 (Santa Cruz, Dallas, TX, USA), anti-TSG101 (Santa Cruz, Dallas, TX, USA), anti-HSP70 (Santa Cruz, Dallas, TX, USA), anti-Calnexin (Proteintech, Chicago, IL, USA), anti-JNK (Abcam, Cambridge, UK), anti-P-JNK (Abcam, Cambridge, UK), anti-CD36 (Proteintech, Chicago, IL, USA), anti-SIRP-α (Proteintech, Chicago, IL, USA), and anti-IL-4R (Proteintech, Chicago, IL, USA) at 4 °C overnight. After washing with a TBST buffer three times, the membrane was incubated with the secondary antibody at room temperature for 90 min. Finally, the protein bands were visualized with enhanced chemiluminescence (Thermo Fisher, Waltham, MA, USA). The immunoreactive bands were quantified with Image J software (Version 1.8.0, National Institutes of Health, Bethesda, MD, USA).

### 2.6. miRNA Sequencing Analysis

The total RNA was extracted from exosomes and endometrial tissues using the miRNeasy^®^Mini Kit (Qiagen, Hilden, Germany) according to instructions. The quality and quantity of the RNA were analyzed using NanoDrop 2000 (Thermo Scientific, Waltham, MA, USA). miRNA sequencings were performed on an Illumina HiSeq2500 (Illumina, San Diego, CA, USA). The differential expression miRNAs were screened with a Fold change (FC) ≥ 1.50 and *p* < 0.05 as the criteria.

### 2.7. Real-Time Quantitative PCR (RT-qPCR) 

The total RNA was reverse transcribed into cDNA with the PrimeScript™ RT reagent Kit (Takara, Shiga, Japan) according to instructions. The real-time quantitative PCR was performed to detect miRNA levels with the TaqMan probe. The 2^−ΔΔCt^ method was used for statistics. The sequence of the primers and probe was as follows: hsa-miR-210-3p-RT: GTCGTATCCAGTGCAGGGTCCGAGGTATTCGCACTGGATACGACTCAGCC; hsa-miR-210-3p-F: CCTGTGCGTGTGACAGC; hsa-miR-210-3p-P: TTCGCACTGGATACGACTCAGCC; U6-RT: AACGCTTCACGAATTTGCGT; U6-S: CTCGCTTCGGCAGCACA; U6-A: AACGCTTCACGAATTTGCGT; U6 probe: AGAAGATTAGCATGGCCCCTGCGCA; qPCR-TYR: GTGCAGGGTCCGAGGT.

### 2.8. Coculture of Exosomes and THP-1-Derived Macrophages

A human acute monocytic leukemia cell line (THP-1) was cultured in an RPMI-1640 medium (Sigma, Saint Louis, MO, USA) with 10% exosome-free fetal bovine serum (SUER, Shanghai, China). Then, THP-1 cells were treated with 100 ng/mL phorbol-12-myristate-13-acetate (PMA) for 48 h to differentiate them into naive macrophages (M0). The typical morphology of macrophages can be observed under a light microscope. Next, exosomes (100 μg/mL) were added and incubated with macrophages. In addition, to examine the effect of the exosomes on macrophages after JNK activation, JNK activator anisomycin (APExBIO, Houston, TX, USA) was coincubated at a concentration of 1 μg/mL for 48 h. Cells were collected for flow cytometry, Western blot, or MAPK Phosphorylation Antibody Arrays.

### 2.9. Uptake of Exosomes by Macrophages

First, the exosomes were stained with PKH67 (Sigma, Saint Louis, MO, USA). Then, the PKH67 green fluorescence-labeled exosomes were co-incubated with macrophages for 6 h. Next, the macrophages were washed with PBS twice and fixed with 4% paraformaldehyde for 30 min. DAPI was added to stain the cell nuclei for 10 min. The uptake of exosomes by macrophages was observed under a fluorescence microscope.

### 2.10. Isolation and Culture of Endometrial Stromal Cells

Endometrial tissues of the control group were collected to isolate and culture endometrial stromal cells. First, the endometrial tissues were rinsed with PBS and cut into little pieces. Next, the tissues were digested with 2.5 mg/mL type I collagenase (Biosharp, Hefei, China) and 0.1 mg/mL DNase I (Biosharp, Hefei, China) for 30 min. The digested tissues were filtered through a 40 μm sieve. Then, the isolated endometrial stromal cells were resuspended in an RPMI-1640 medium with a 10% fetal bovine serum.

### 2.11. Transwell Migration and Invasion Assay

The endometrial stromal cells were starved with the serum-free RPMI-1640 medium for 24 h. In the transwell migration assay, 5 × 10^4^ cells suspended in a 200 μL serum-free RPMI-1640 medium were seeded in the upper chamber of the transwell chamber. The lower chambers were THP-1-derived macrophages, which were treated with exosomes (100 μg/mL) and cultured in an RPMI-1640 medium with a 10% exosome-free fetal bovine serum. After 48 h, the transwell chambers were removed and washed twice with PBS. Then, the migrated cells were fixed with methanol for 10 min and stained with 0.1% crystal violet for 30 min. The number of migrated cells were counted under the microscope. Cells in five fields per chamber were randomly selected for counting. In the transwell invasion assay, 1 × 10^5^ endometrial stromal cells were seeded in the upper chamber of the transwell chamber, and the chamber bottom was precoated with Matrigel. The next experimental operations were consistent with the transwell migration assay.

### 2.12. miRNA Mimics or Inhibitor Transfectio

THP-1-derived macrophages were cultured in an RPMI-1640 medium (Sigma, Saint Louis, MO, USA) with a 10% exosome-free fetal bovine serum (SUER, Shanghai, China). Then, miR-210-3p mimics, mimics negative control (NC), miR-210-3p inhibitor, and inhibitor negative control (RiBoBio, Guangzhou, China) were used to transfect macrophages with Lipofectamine 2000 (Invitrogen, Carlsbad, CA, USA). The transfection efficiencies were detected using RT-qPCR. After 48 h of transfection, the macrophages were collected for flow cytometry.

### 2.13. MAPK Phosphorylation Antibody Arrays

A Human MAPK Phosphorylation Antibody Array (ab211061, Abcam, Cambridge, UK) was used to detect the phosphorylation of the MAPK pathway in macrophages after exosome intervention. In brief, macrophages were washed with PBS twice after incubating the exosomes with macrophages for 48 h. Then, a cell lysis buffer was added to the lysate macrophages at 4 °C. After 30 min of lysis, the supernatant was collected by centrifugation at 14,000× *g* for 10 min. The total protein concentration of the supernatant was detected using the BCA Protein Assay kit (Beyotime, Shanghai, China). The supernatant was diluted with a cell lysis buffer to ensure that the total protein concentration of each cell lysis supernatant was 80 μg/mL. Then, the cell lysates were incubated in 2 mL of blocking buffer at room temperature for 30 min. Next, the blocking buffer was removed, and the protein samples were incubated overnight at 4 °C. After being washed three times with wash buffer I and wash buffer II, the membranes were incubated with a detection antibody cocktail for 120 min at room temperature. Then, the membranes were incubated with horseradish peroxidase-conjugated antirabbit IgG for 120 min at room temperature. The membranes were visualized with the chemiluminescence imaging system (Bio-Rad, Hercules, CA, USA).

### 2.14. Flow Cytometry

The digested macrophages were washed twice with PBS. Then, the cells were centrifuged at 1500 rpm for 5 min to remove the supernatant, and then they were incubated with 5 μL of anti-CD80-FITC (eBioscience, San Diego, CA, USA) at 4 °C for 30 min in the dark. Flow cytometry was performed using a CytoFLEX flow cytometer (Beckman Coulter, Brea, CA, USA), and the cells were analyzed using CytExpert software (Version 2.4, Beckman Coulter, Brea, CA, USA).

### 2.15. Statistical Analysis

Statistical analyses were performed using GraphPad Prism 7.0 software (GraphPad Software Inc, La Jolla, CA, USA). The data are shown as the mean and standard deviation, and *p* < 0.05 is considered to be statistically significant. The student’s *t*-test was used for statistical analysis between two unpaired groups. A one-way analysis of variance (ANOVA) was used to compare differences between three or more groups.

## 3. Result

### 3.1. Identification of Exosomes

The morphology of exosomes was observed using TEM. Exosomes showed a typical round and cup-shaped morphology (Figure 1A). Moreover, NTA showed that the main peak of the exosomes was 111.7 nm and the proportion of the main peak was 94.1% (Figure 1B). Exosome markers were verified using a Western blot. The results showed that exosome positive markers (CD63, TSG101, and HSP70) were expressed in isolated exosomes. Meanwhile, exosome negative markers (Calnexin) were not expressed in our isolated exosomes (Figure 1C). These characteristics indicated that we successfully isolated exosomes from the uterine aspirate fluid.

### 3.2. Effect of Exosomes on the Polarization of Macrophages

PKH67 green-fluorescence-labeled exosomes and DAPI stained the macrophage nuclei (Figure 2A,B,D,E). After incubating the exosomes with macrophages for 6 h, the uptake of exosomes by macrophages was observed under a fluorescence microscope. The results showed that exosomes of the uterine aspirate fluid were taken up by macrophages in both the EMs group and the control group (Figure 2C,F).

After incubating exosomes of the uterine aspirate fluid with macrophages for 48 h, flow cytometry was used to detect the proportion of CD80^+^ macrophages (M1 macrophages). The results revealed that exosomes of the uterine aspirate fluid in patients with endometriosis decreased the proportion of CD80^+^ macrophages compared with the control and PBS (Figure 2G–J).

### 3.3. Effects of Exosomes on Macrophage Receptors

The expression of CD36 was decreased after interfering with exosomes of the uterine aspirate fluid in patients with endometriosis, while the expression of SIRP-α was not affected. Additionally, the expression of IL-4R was increased in the EMs group compared to the control (Figure 3).

### 3.4. Exosomes Affect Macrophage Polarization through the MAPK Pathway

MAPK Phosphorylation Antibody Arrays were used to detect changes in the MAPK pathway phosphorylated proteins in macrophages after intervention with exosomes of the uterine aspirate fluid. It was found that compared with the control group, the expression of the phosphorylated proteins JNK, MKK3, MKK6, P38, P53, P70S6k, and RSK1 was significantly decreased in the EMs group (Figure 4). In addition, western blot further confirmed that the expression of P-JNK in macrophages was decreased after intervention with exosomes of endometriosis compared with the control and PBS (Figure 5A,B). However, the JNK activator anisomycin [21] could increase P-JNK expression (Figure 5A,B). Exosomes from endometriosis could decrease the proportion of CD80^+^ macrophages (Figure 2G,H) whereas treatment with anisomycin could increase the CD80^+^ macrophage level (Figure 5C,F).

### 3.5. Effect of Exosomes on Migration and Invasion of Endometrial Stromal Cells

The transwell experiment was performed to investigate whether exosomes could affect endometrial stromal cells by affecting macrophages. The upper chambers of the transwell chamber were endometrial stromal cells. The lower chambers were macrophages treated with exosomes. The result shows that compared with the control group, exosomes from the EMs group could promote the migration and invasion of endometrial stromal cells by acting on macrophages (Figure 6).

### 3.6. Differentially Expressed miRNAs in Exosomes of the Uterine Aspirate Fluid

miRNA sequencing analysis was performed in exosomes derived from the uterine aspirate fluid between the EMs group and the control group. A total of 1357 exosomal miRNAs were detected in the two groups. Differential expression exosomal miRNAs were screened with a Fold change (FC) ≥ 1.50 and *p* < 0.05 as criteria. Seven upregulated exosomal miRNAs were screened from exosomes of the EMs group compared with the control group: miR-210-3p, miR-20b-5p, miR-625-5p, miR-342-5p, miR-155-5p, miR-146A-5p, and miR-130b-3p. Meanwhile, two downregulated exosomal miRNAs were screened: miR-335-3p and miR-132-5p were observed in endometriosis (Figure 7A,B). RT-qPCR further verified the significant differentially expressed miRNA in miRNA sequencing. The results showed that the expression of exosomal miR-210-3p was elevated in endometriosis compared with the control no matter in the proliferative phase or the secretory phase (Figure 7C).

### 3.7. Differentially Expressed miRNAs in Endometrial Tissues

Similarly, differential expression miRNAs were screened with a Fold change (FC) ≥ 1.50 and *p* < 0.05 as criteria. A total of 1327 miRNAs were detected in the EMs group and the control group. Eighteen upregulated miRNAs (miR-210-3p, miR-34c-5p, miR-375-3p, etc.) and four downregulated miRNAs (miR-708-5p, miR-143-5p, and miR-132-3p) were found in the eutopic endometrium from patients with endometriosis compared with the control (Figure 8A,B). miR-210-3p expression was increased in the eutopic endometrium of patients with endometriosis no matter in the proliferative phase or the secretory phase (Figure 8C). The results were consistent with miRNA sequencing. Among them, only miR-210-3p was expressed consistently both in the eutopic endometrium and exosomes in endometriosis. The expression of miR-210-3p was increased in both exosomes and the eutopic endometrium in endometriosis.

### 3.8. Effect of miR-210-3p on the Polarization of Macrophages

The relative expression of miR-210-3p was detected by RT-qPCR to verify the transfection efficiency. miR-210-3p mimics could increase the expression of miR-210-3p in macrophages after transfection, indicating successful transfection (Figure 9A). The proportion of CD80^+^ macrophages was significantly decreased after transfection with miR-210-3p mimics compared with the PBS and miR-210-3p mimics-NC (Figure 9B–E), while the proportion of CD80^+^ macrophages was increased after transfection with the miR-210-3p inhibitor (Figure 9F).

## 4. Discussion

At present, the study of exosomes in endometriosis mainly focuses on blood, peritoneal fluid, and endometrial stromal cells [22,23,24,25]. There are few studies on exosomes of uterine aspirate fluid. We successfully isolated exosomes from the uterine aspirate fluid. This study is the first to investigate the pathogenesis of endometriosis from exosomes of the uterine aspirate fluid.

In our study, we found that the proportion of CD80^+^ macrophages decreased after macrophages were treated with exosomes of the uterine aspirate fluid from patients with endometriosis. CD80 is a surface marker of M1 macrophages [26]. Macrophages are plastic cell populations, which can be divided into classically activated M1 macrophages and alternatively activated M2 macrophages [26]. Studies have shown that the reduction in M1 macrophages could help ectopic lesions escape immune surveillance, thus promoting the development of endometriosis [27]. Additionally, the predominance of M2 macrophages in the abdominal cavity is associated with endometriosis [28,29]. According to our findings, exosomes from the uterine cavity in patients with endometriosis reduced M1 macrophages and promoted the migration and invasion of endometrial stromal cells by acting on macrophages. Therefore, we speculated that exosomes from the uterine cavity in endometriosis promoted the formation of an M2 macrophage-dominated immunological microenvironment in the peritoneal cavity, favoring the development of ectopic lesions.

In addition, we found that exosomes from the uterine cavity in endometriosis could decrease the proportion of CD80^+^ macrophages by inhibiting the JNK pathway. The MAPK signaling pathways are important for regulating macrophage function and plasticity [30]. MAPKs are evolutionarily conserved serine/threonine kinases that can participate in cell proliferation, differentiation, and apoptosis [31,32]. JNK is a key protein kinase in the MAPK pathway cascade, which can regulate macrophage phenotypes [30]. In our study, we found that exosomes from endometriosis suppressed the phosphorylation of JNK in macrophages. Furthermore, the ability of exosomes from endometriosis to decrease the proportion of M1 macrophages could be reversed by the JNK activator anisomycin. This suggests that exosomes from the uterine cavity in endometriosis could inhibit macrophage M1 polarization by inhibiting the JNK pathway, and this effect can be treated with anisomycin. Zha et al. demonstrated that the activation of the JNK signaling pathway enhances macrophage M1 polarization [33]. Chen et al. also found that the inhibition of JNK phosphorylation was linked to decreased M1 macrophages, which was consistent with our results [34].

We found that miR-210-3p was the only miRNA with a consistent differential expression between exosomes and eutopic endometrial tissues in endometriosis. The types of miRNAs carried by exosomes did not match endometrial tissues. This is mainly related to the specific miRNA sorting mechanism involved in the formation of exosomes. Some studies have shown that the mechanism of sorting specific miRNAs into exosomes was related to RNA-binding proteins such as hnRNPA2B1 and Y-box protein I (YBX1) [35,36]. RNA-binding proteins bind a specific subset of miRNAs and load them into exosomes. Cha et al. demonstrated that gene mutations were also associated with selective miRNA sorting into exosomes [37]. The mechanism of sorting specific miRNAs from eutopic endometrial tissues into exosomes of the uterine aspirate fluid needs to be further studied.

Furthermore, our study showed that miR-210-3p mimics inhibit the M1 polarization of macrophages. This indicates that the exosomes from endometriosis that inhibit macrophage M1 polarization by inhibiting the JNK pathway may be closely related to exosomal miR-210-3p. Dai et al. discovered that the elevation of miR-210-3p can decrease the expression of BARD1, which contributes to the development of endometriosis [38]. The exosomes from the uterine cavity in endometriosis were highly expressed miR-210-3p. These exosomes could inhibit macrophage M1 polarization after entering the peritoneal cavity by inhibiting the activation of the JNK signaling pathway. The decrease in M1 macrophages in the peritoneal cavity allowed the ectopic lesions to escape immune surveillance and be successfully implanted.

Exosomes in endometriosis not only affect the phenotype of macrophages but also affect the function of macrophages by affecting the expression of macrophage receptors. The expression of the macrophage receptor CD36 was downregulated after exosomes of the uterine aspirate fluid were taken from the endometriosis intervention. CD36 belongs to the Class B scavenger receptor family, which is closely related to the phagocytosis of macrophages [39]. Chuang et al. found that the expression of CD36 in the peritoneal macrophages of patients with endometriosis decreased [40]. We speculate that exosomes from the uterine cavity in endometriosis may affect the phagocytosis of macrophages to ectopic lesions by downregulating the expression of CD36 in peritoneal macrophages. Additionally, our findings revealed that the expression of the macrophage receptor IL-4R was upregulated after obtaining the exosome of the endometriosis intervention. IL-4R is a cytokine receptor associated with alternatively activated M2 macrophages [41,42]. A limitation of our study is that more research is needed on how macrophage receptors affects the pathogenesis of endometriosis after exosome intervention.

## 5. Conclusions

Exosomes were successfully isolated from uterine aspirate fluid. Exosomes from the uterine cavity were highly expressed miR-210-3p in patients with endometriosis. These exosomes could transmit signals from the uterine cavity to the abdominal cavity. Then, it could inhibit the M1 polarization of macrophages by inhibiting the activation of the JNK signaling pathway. The reduction in M1 macrophages in the peritoneal cavity could promote the migration and invasion of ectopic endometrial stromal cells (Figure 10). Exosomes also affect macrophage function by affecting macrophage receptors such as CD36 and IL-4R. To sum up, exosomes from the uterine cavity in patients with endometriosis may affect the phenotype of macrophages by inhibiting the JNK signaling pathway, hence facilitating the formation of a peritoneal immunological microenvironment favorable for the development of endometriosis.

## Figures and Tables

**Figure 1 biomedicines-10-03110-f001:**
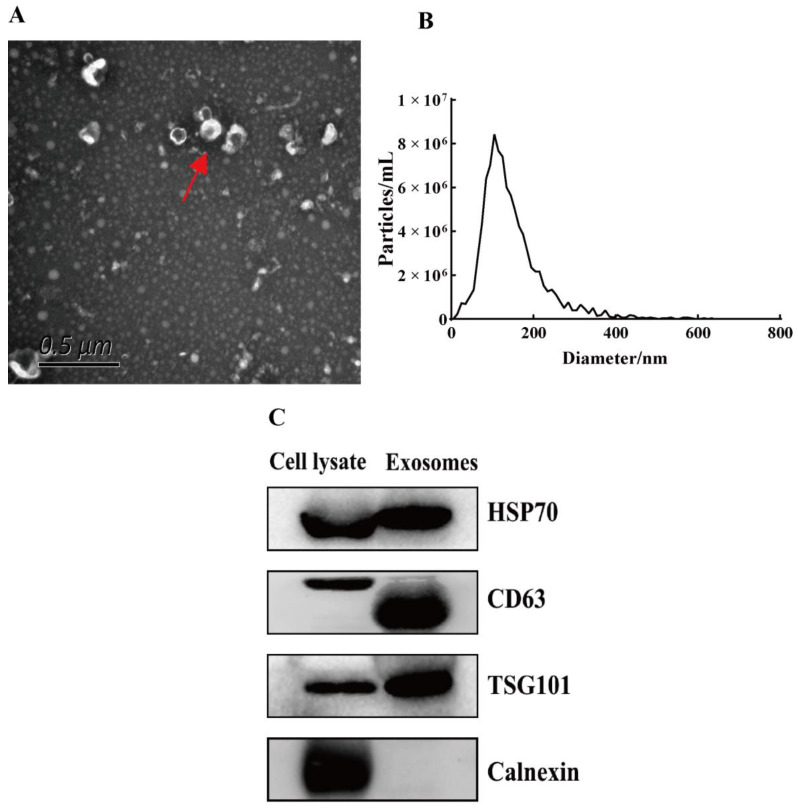
Identification of exosomes. (**A**) Exosomes (red arrow) were observed by transmission electron microscopy. Scale bar: 0.5 μm. (**B**) The particle size of exosomes was analyzed with NTA. (**C**) Exosome positive markers (CD63, TSG101, and HSP70) and negative marker (Calnexin) were verified by Western blot.

**Figure 2 biomedicines-10-03110-f002:**
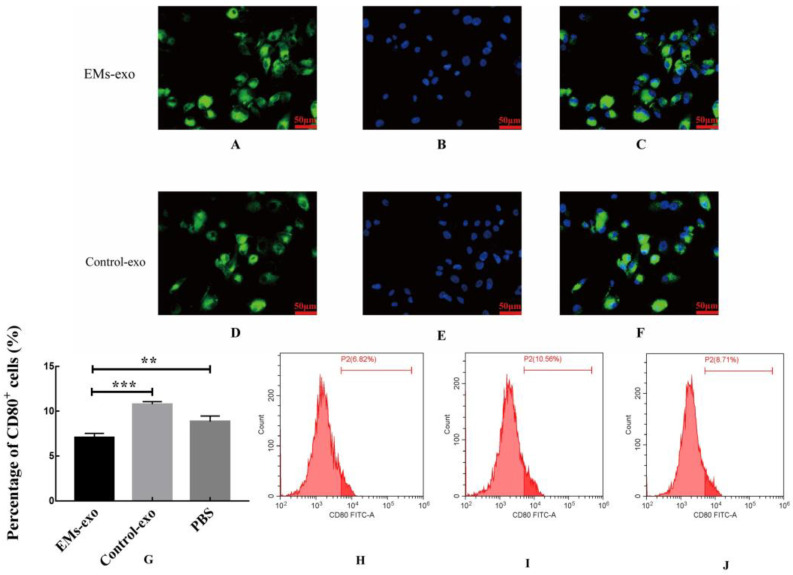
Exosomes of the uterine aspirate fluid in patients with endometriosis decreased the proportion of CD80^+^ macrophages. (**A**) Exosomes from the EMs group were labeled with green fluorescence. (**B**) The macrophage nuclei were stained blue. (**C**) The uptake of exosomes from the EMs group by macrophages. (**D**) Exosomes from the control group were labeled with green fluorescence. (**E**) The macrophage nuclei were stained blue. (**F**) The uptake of exosomes from the control group by macrophages. Scale bar: 50 μm. (**G**) The flow cytometry was used to detect the proportion of CD80^+^ macrophages. (**H**) The proportion of CD80^+^ macrophages after intervention with exosomes from the EMs group. (**I**) The proportion of CD80^+^ macrophages after intervention with exosomes from the control group. (**J**) The proportion of CD80^+^ macrophages after intervention with PBS. Data are expressed as the mean ± standard error of the mean (SEM) (** *p* < 0.01, *** *p* < 0.001).

**Figure 3 biomedicines-10-03110-f003:**
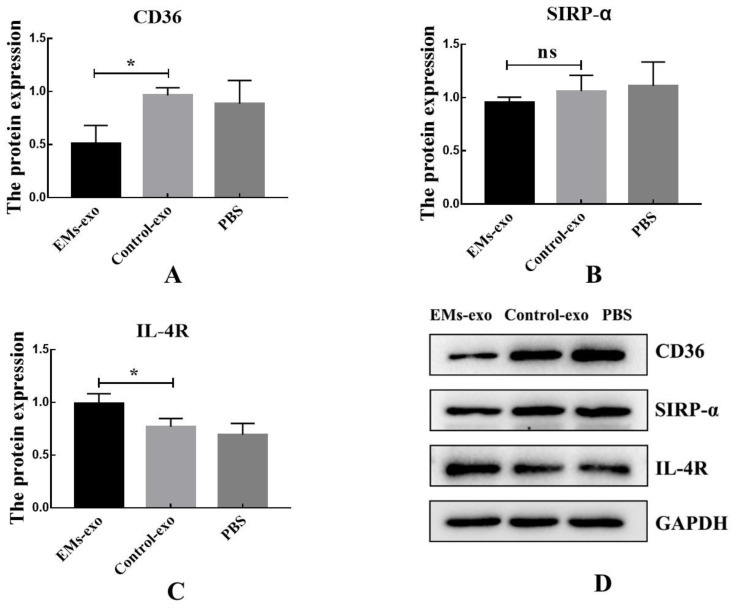
Exosomes influence the expression of macrophage receptors. (**A**) The protein expression of macrophage receptors CD36. (**B**) The protein expression of macrophage receptors SIRP-α. (**C**) The protein expression of macrophage receptors IL-4R. (**D**) Western blot was used to detect the protein expression of macrophage receptors after the intervention of exosomes from the EMs group, exosomes from the control group, and PBS. Data are expressed as the mean ± SEM (* *p* < 0.05). ns: not significant.

**Figure 4 biomedicines-10-03110-f004:**
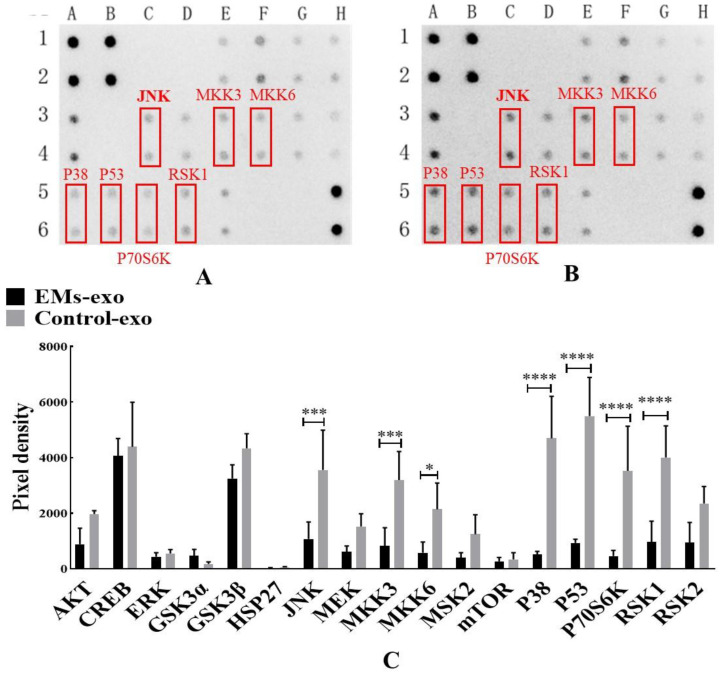
Exosomes of the uterine aspirate fluid influence the MAPK pathway phosphorylated proteins in macrophages. (**A**) The expression of MAPK pathway phosphorylated proteins after intervention with exosomes from the EMs group. (**B**) The expression of MAPK pathway phosphorylated proteins after intervention with exosomes from the control group. (**C**) The differentially expressed MAPK pathway phosphorylated proteins. The red box represents the differentially expressed proteins compared with the control. Data are expressed as the mean ± SEM (* *p* < 0.05, *** *p* < 0.001, **** *p* < 0.0001).

**Figure 5 biomedicines-10-03110-f005:**
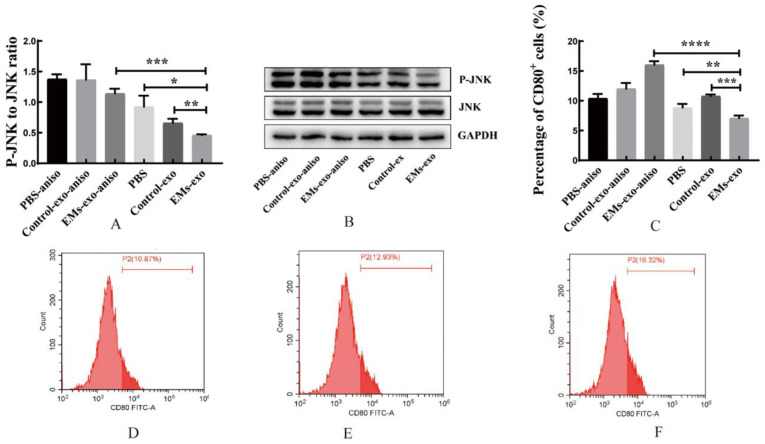
Exosomes of the EMs group decreased the expression of P-JNK in macrophages. (**A**,**B**) Western blot was used to verify the expression of P-JNK in macrophages. (**C**) The flow cytometry was used to detect the proportion of CD80^+^ macrophages. (**D**) The effects of PBS on the proportion of CD80^+^ macrophages after adding anisomycin. (**E**) After adding anisomycin, the effects of exosomes from the control group on the proportion of CD80^+^ macrophages. (**F**) After adding anisomycin, the effects of exosomes from the EMs group on the proportion of CD80^+^ macrophages. JNK activator: anisomycin (aniso). Data are expressed as the mean ± SEM (* *p* < 0.05, ** *p* < 0.01, *** *p* < 0.001, **** *p* < 0.0001).

**Figure 6 biomedicines-10-03110-f006:**
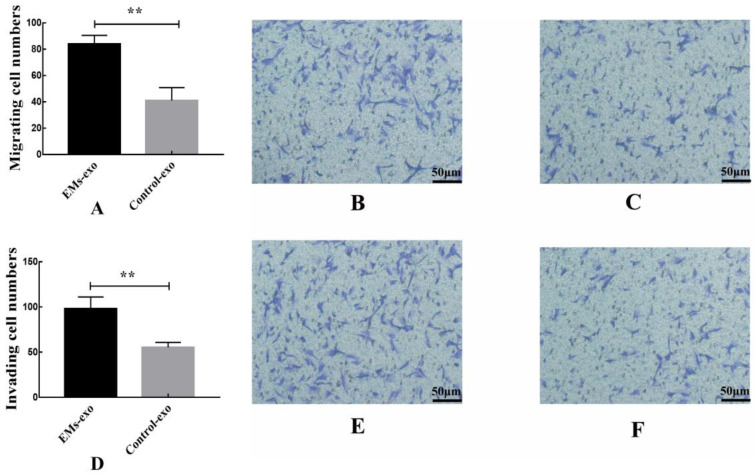
Exosomes from the EMs group could promote the migration and invasion of endometrial stromal cells by acting on macrophages. (**A**) The migration of endometrial stromal cells. (**B**) The effects of exosomes from the EMs group on the migration of endometrial stromal cells after acting on macrophages. (**C**) The effects of exosomes from the control group on the migration of endometrial stromal cells after acting on macrophages. (**D**) The invasion of endometrial stromal cells. (**E**) The effects of exosomes from the EMs group on the invasion of endometrial stromal cells after acting on macrophages. (**F**) The effects of exosomes from the control group on the invasion of endometrial stromal cells after acting on macrophages. Data are expressed as the mean ± SEM (** *p* < 0.01). Scale bar: 50 μm.

**Figure 7 biomedicines-10-03110-f007:**
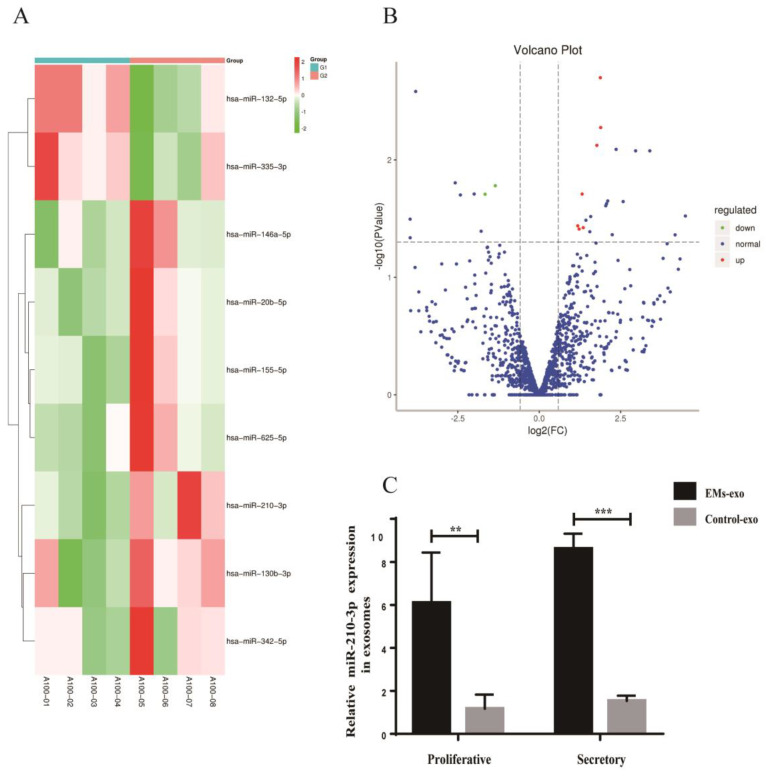
Differentially expressed miRNAs in exosomes of the uterine aspirate fluid. (**A**) Heatmap of the differentially expressed miRNAs in exosomes from the EMs group compared with the control. (**B**) The Volcano plot of the differentially expressed miRNAs in the exosomes from the EMs group compared with the control. The red color indicates increased expression while the green color indicates decreased expression. A100−01 to A100−04 were the control group; A100−05 to A100−08 were the EMs group. (**C**) RT-qPCR was used to verify the expression of miR-210-3p in exosomes from the EMs group and the control group. Data are expressed as the mean ± SEM (** *p* < 0.01, *** *p* < 0.001).

**Figure 8 biomedicines-10-03110-f008:**
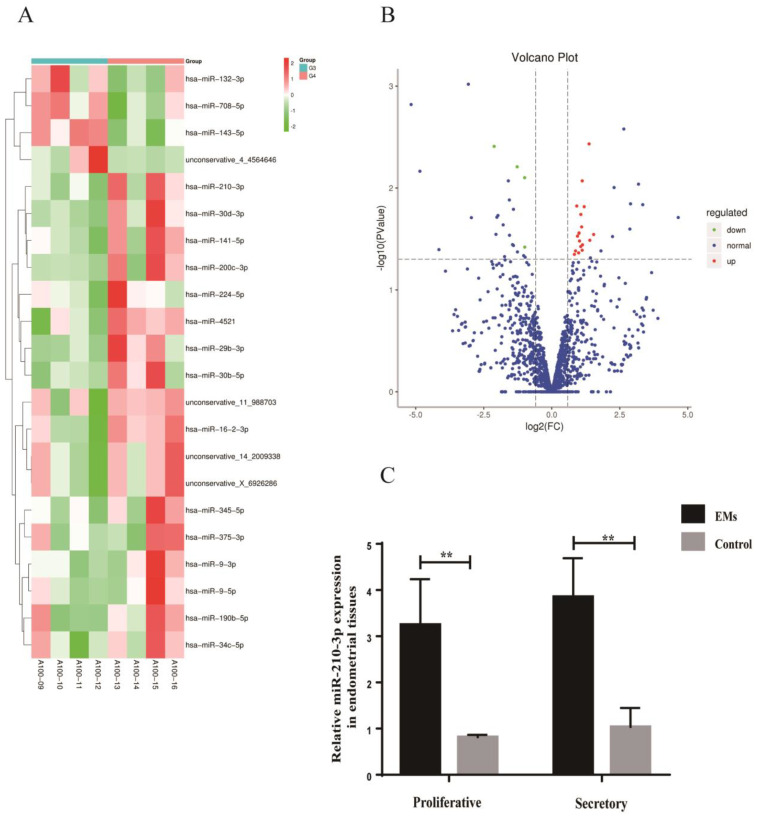
Differentially expressed miRNAs in endometrial tissues. (**A**) Heatmap of the differentially expressed miRNAs in endometrial tissues from patients with endometriosis compared with the control. (**B**) The Volcano plot of the differentially expressed miRNAs in endometrial tissues from patients with endometriosis compared with the control. The red color indicates increased expression while the green color indicates decreased expression. A100−09 to A100−12 were the control group; A100−13 to A100−16 were the EMs group. (**C**) RT-qPCR was used to verify the expression of miR-210-3p in endometrial tissues from the EMs group and the control group. Data are expressed as the mean ± SEM (** *p* < 0.01).

**Figure 9 biomedicines-10-03110-f009:**
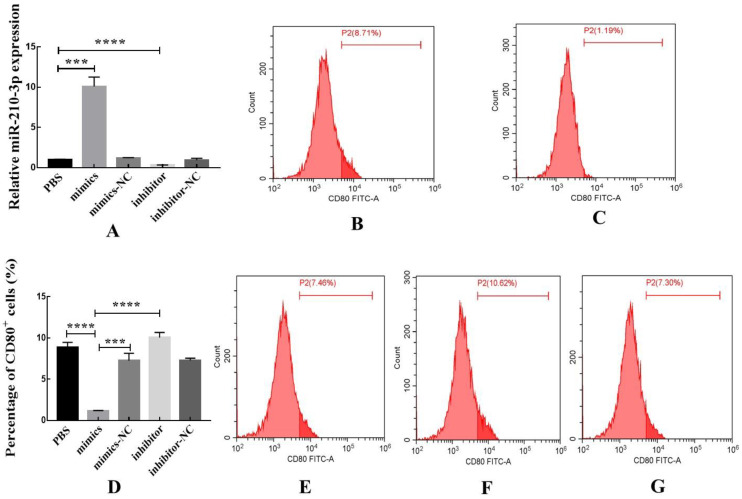
miR-210-3p could decrease the proportion of CD80^+^ macrophages. (**A**) The transfection efficiency was verified by RT-qPCR. (**B**) The effects of PBS on the proportion of CD80^+^ macrophages. (**C**) The effects of miR-210-3p mimics on the proportion of CD80^+^ macrophages. (**D**) The flow cytometry was used to detect the proportion of CD80^+^ macrophages. (**E**) The effects of miR-210-3p mimics-NC on the proportion of CD80^+^ macrophages. (**F**) The effects of miR-210-3p inhibitor on the proportion of CD80^+^ macrophages. (**G**) The effects of miR-210-3p inhibitor-NC on the proportion of CD80^+^ macrophages. Data are expressed as the mean ± SEM (*** *p* < 0.0001, **** *p* < 0.0001).

**Figure 10 biomedicines-10-03110-f010:**
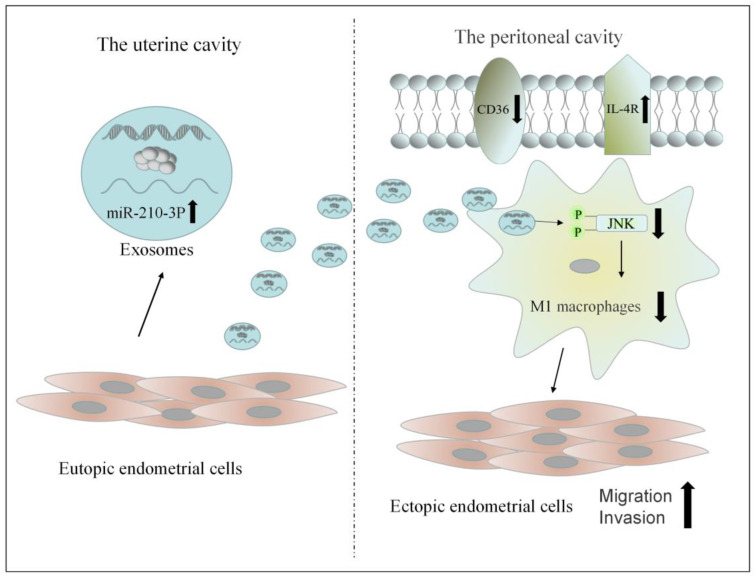
The possible mechanism of exosomes from the uterine aspirate fluid involved in the pathogenesis of endometriosis.

## Data Availability

The data presented in this study are available from the corresponding author upon reasonable request.

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
