# Peer review of "Exosomes from the Uterine Cavity Mediate Immune Dysregulation via Inhibiting the JNK Signal Pathway in Endometriosis"

_biomedicines, 2022, doi:10.3390/biomedicines10123110_

Round 1
Reviewer 1 Report
The study by Jiang and colleagues investigates the role of uterine cavity exosomes in the pathogenesis of endometriosis. In particular, authors show that these exosomes contribute to the formation of an immunological microenvironment predisposing for endometriosis development through affection of macrophages phenotype, including their polarization, via desensitization of JNK signaling pathway.
The study is very interesting and original. It is clearly written and comprehensive. Methods are appropriate and results well presented. Results support conclusions.
English is fine.
Author Response
Response:Thank you very much for your positive comment on our manuscript.
Reviewer 2 Report
In this study, the authors examined the effects of exosomes from the uterine cavity of endometriosis patients on population and intracellular signaling in macrophages and migration of endometrial stromal cells. Furthermore, they found the several miRNAs differentially existed in exosomes and expressed in endometrial tissues from endometriosis patients such as miR-210-2p. The research has some degree of scientific significance of the role of exosomes on endometriosis. However, the figures and results are not well presented.
Major comments,
1. There are many reports that showed the involvement of exosomes on endometriosis. The author should describe what is novel findings in the manuscript.
2. How many samples (patients) of endometriosis and the control were used in this study? This should be clearly described in the “2.1. Sample collection”.
3. The patient's background should also be described in the Materials and Methods section.
4. How many times did the author repeat each experiment?
5. All figure titles are inappropriate and missing.
6. The author clearly describes why they examined the effects of exosomes on the expression of CD36, SIRP-a, and IL-4R in the macrophage-like cells in the results section, but not in the discussion, to easy to understand the meaning (page 7 line258).
7. How and what concentration of anisomycin was used in the study? There is no information on the vendor, concentration, and experimental schedules. These should be added in the Materials and Methods or Figure legend.
8. In figure 5A and B, anisomycin did not repress phosphorylation JNK.
9. Did the author examine the effects of exosomes on the proliferation of endometrial stromal cells? The data of the proliferation assay strengthens your conclusion that EM-exosome specifically stimulates the migration and invasion of endometrial stromal cells.
10. Although miR-210-3p mimic downregulated the ratio of CD 80 positive cells, does the mimic also repress the phosphorylation of JNK and the expression of CD36 and increase the expression of IL-4R?
Author Response
- There are many reports that showed the involvement of exosomes on endometriosis. The author should describe what is novel findings in the manuscript.
Response 1: Thanks for your thoughtful suggestion. Many reports have shown the role of exosomes in endometriosis. However, in these studies, exosomes were mainly isolated from blood, peritoneal fluid, and endometrial stromal cells. As we know, it is the first time investigate the pathogenesis of endometriosis from exosomes of the uterine aspirate fluid. Our research is more consistent with the theory of retrograde menstruation. We also described it in the the first paragraph of discussion. At present, the study of exosomes in endometriosis mainly focuses on blood, peritoneal fluid, and endometrial stromal cells. Exosomes from the uterine aspirate fluid can reflect the physiologic state of the uterine cavity, which is a novel biomarker for the study of gynecological diseases. Exosomes were successfully isolated from the uterine aspirate fluid. It is the first time to investigate the pathogenesis of endometriosis from exosomes of the uterine aspirate fluid (page 13, Lines 362-367).
- How many samples (patients) of endometriosis and the control were used in this study? This should be clearly described in the “2.1. Sample collection”.
Response 2: We have added it to the revised manuscript. The EMs group (n = 22) was the patients who underwent laparoscopic surgery for ovarian endometriosis (Lines 77). The control group (n = 25) was the patients who underwent surgical treatment for other benign gynecological diseases except for endometriosis (page 2, Lines 82).
- The patient's background should also be described in the Materials and Methods section.
Response 3: We have added it to the revised manuscript (page 2, Lines 78-85). The EMs group (n = 22) was the patients who underwent laparoscopic surgery for ovarian endometriosis at the Second Xiangya Hospital of Central South University between January 2020 and August 2021. The average age of the EMs group was 29.32 years. All the patients with endometriosis were at stage III/IV according to the revised American Society for Reproductive Medicine classification system. The control group (n = 25) was the patients who underwent surgical treatment for other benign gynecological diseases except for endometriosis, such as simple ovarian cyst, uterine fibroids, and tubal ligation. The average age of the control group was 28.44 years. All patients had signed informed consent. All patients included had a regular menstrual cycle and had not taken any hormone drugs or been placed on an intrauterine device for at least 6 months before surgery.
- How many times did the author repeat each experiment?
Response 4: Each experiment was independently repeated three times.
- All figure titles are inappropriate and missing.
Response 5: Thanks for your thoughtful suggestion. All figure titles have been corrected in the revised manuscript.
- The author clearly describes why they examined the effects of exosomes on the expression of CD36, SIRP-a, and IL-4R in the macrophage-like cells in the results section, but not in the discussion, to easy to understand the meaning (page 7 line258).
Response 6: Thanks for raising this issue. Correction has been made in the revised manuscript (page 7, line258). The expression of CD36 was decreased after interfering with exosomes of the uterine aspirate fluid in patients with endometriosis, while the expression of SIRP-α was not affected. Besides, the expression of IL-4R was increased in the EMs group compared to the control.
- How and what concentration of anisomycin was used in the study? There is no information on the vendor, concentration, and experimental schedules. These should be added in the Materials and Methods or Figure legend.
Response 7: Thanks for raising this critical issue. Accordingly, several sentences have been added in the Methods (Page 4, Lines 158-160) of the revised manuscript. Exosomes (100μg/mL) were added and incubated with macrophages. In addition, to examine the effect of exosomes on macrophages after JNK activation, JNK activator anisomycin (APExBIO, USA) was co-incubated at a concentration of 1μg/mL for 48 h.
- In figure 5A and B, anisomycin did not repress phosphorylation JNK.
Response 8: Because anisomycin was a JNK agonist, it did not inhibit JNK phosphorylation. In our study, we found that exosomes from the uterine aspirate fluid in endometriosis could inhibit macrophage M1 polarization by inhibiting the phosphorylation of JNK. Therefore, we subsequently added JNK agonists anisomycin to investigate whether this phenomenon could be reversed by anisomycin.
- Did the author examine the effects of exosomes on the proliferation of endometrial stromal cells? The data of the proliferation assay strengthens your conclusion that EM-exosome specifically stimulates the migration and invasion of endometrial stromal cells.
Response 9: We didn't. Because we mainly want to study whether exosomes participate in the development of endometriosis by mediating immune dysregulation. Macrophages play a crucial role in the pathogenesis of endometriosis, so we chose macrophages for further study. In addition, previous studies have shown that EMS-exo could promote endometrial stromal cell proliferation, migration, and invasion. References are attached below.
- Liu Ting, Liu Mei, Zheng Caihua et al. Exosomal lncRNA CHL1-AS1 Derived from Peritoneal Macrophages Promotes the Progression of Endometriosis via the miR-610/MDM2 Axis. [J]. Int J Nanomedicine, 2021, 16: 5451-5464.
- Qiu Jun-Jun, Lin Ying-Ying, Tang Xiao-Yan et al. Extracellular vesicle-mediated transfer of the lncRNA-TC0101441 promotes endometriosis migration/invasion. [J]. Exp Cell Res, 2020, 388: 111815.
- Although miR-210-3p mimic downregulated the ratio of CD 80 positive cells, does the mimic also repress the phosphorylation of JNK and the expression of CD36 and increase the expression of IL-4R?
Response 10: Thanks for your comment on the present study. In this study, we mainly want to elaborate on the role of exosomes, the role of exosomal miR-210-3p will be further studied in the future.

Reviewer 3 Report
The purpose of Jiang et al. was to investigate the role of exosomes in pathogenesis of endometriosis. This is very interesting paper, however I have only few questions:
1. In the abstract, “exosomes from endometriosis” statement is too general and should be made more specific.
2. There is no typical structure in the abstract. The authors made no mention of the materials or the methods used in this experiment.
3. The introduction is too extensive. Especially the first paragraphs, where the authors present general information about endometriosis and theories of the emergence of endometriosis, seem to be too long.
4. Could the authors add more information about the patients?
5. What was the original purpose of this procedure?
6. What was the menstrual cycle phase? Could this affect the results obtained?
7. With what type of endometriosis? Have any patients diagnosed with adenomyosis?
8. Point 2.10 – is that mean that the culture endometrial stromal cells were isolated only from control group?
9. The discussion is too extensive. It should be more concerned with discussing the results in the context of other studies, eg the whole excerpt about M1M2 macrophages can be replaced with one sentence.
10. I also miss a clear connection between the parameters studied. Each paragraph is as if separate. It would be nice to present the relationships between them.
Author Response
- In the abstract, “exosomes from endometriosis” statement is too general and should be made more specific.
Response 1: Thank you for your insightful suggestion. We have corrected it to "exosomes of the uterine aspirate fluid from endometriosis".
- There is no typical structure in the abstract. The authors made no mention of the materials or the methods used in this experiment.
Response 2: Several sentences have been added to the abstract of the revised manuscript (page 1, Lines 11-17). Exosomes of the uterine aspirate fluid were isolated and co-cultured with macrophages for 48h. Flow cytometry was used to detect macrophage polarization. Human MAPK Phosphorylation Antibody Array and western blot were used to detect the phosphorylation of the MAPK pathway. microRNA sequencing analysis was used to detect differentially expressed miRNAs. Our research found that exosomes of the uterine aspirate fluid from endometriosis can reduce the proportion of CD80+ macrophages. Meanwhile, it could inhibit the expression of P-JNK in macrophages.
- The introduction is too extensive. Especially the first paragraphs, where the authors present general information about endometriosis and theories of the emergence of endometriosis, seem to be too long.
Response 3: We have condensed and deleted some sentences in the introduction (pages 1-2, Lines 31-33, 37-46, 51, 58-59, 62)
- Could the authors add more information about the patients?
Response 3: We have added it to the revised manuscript (page 2, Lines 78-85). The EMs group (n = 22) was the patients who underwent laparoscopic surgery for ovarian endometriosis at the Second Xiangya Hospital of Central South University between January 2020 and August 2021. The average age of the EMs group was 29.32 years. All the patients with endometriosis were at stage III/IV according to the revised American Society for Reproductive Medicine classification system. The control group (n = 25) was the patients who underwent surgical treatment for other benign gynecological diseases except for endometriosis, such as simple ovarian cyst, uterine fibroids, and tubal ligation. The average age of the control group was 28.44 years.
- What was the original purpose of this procedure?
Response 5: At present, the study of exosomes in endometriosis mainly focuses on blood, peritoneal fluid, and endometrial stromal cells. Exosomes from the uterine aspirate fluid can reflect the physiologic state of the uterine cavity, which is a novel biomarker for the study of gynecological diseases. However, the role of exosomes from the uterine aspirate fluid in endometriosis is rarely reported, so our original purpose was to investigate whether exosomes from the uterine cavity could participate in the development of endometriosis.
- What was the menstrual cycle phase? Could this affect the results obtained?
Response 6: Patients at the proliferative phase and the secretory phase were included. This couldn't affect the results obtained. We compared the protein and RNA concentrations of exosomes between patients at the proliferative phase and the secretory phase. There was no statistical difference.
- With what type of endometriosis? Have any patients diagnosed with adenomyosis?
Response 7: All the patients of the EMs group included in our study were ovarian endometriosis without adenomyosis. The diagnosis was confirmed by laparoscopic surgery and histopathology.
- Point 2.10 – is that mean that the culture endometrial stromal cells were isolated only from control group?
Response 8: Yes. Because in this part of the experiment, our variable was exosomes from the EMs group and the control group. We wanted to compare whether exosomes from the EMs group and the control group could influence the migration and invasion of endometrial stromal cells by acting on macrophages. Therefore, to control variables, we isolated endometrial stromal cells only from the control group.
- The discussion is too extensive. It should be more concerned with discussing the results in the context of other studies, eg the whole excerpt about M1M2 macrophages can be replaced with one sentence.
Response 9: We have condensed and deleted some sentences in the discussion (Pages 13-14, Lines 368, 372, 386-387, 393-395, 402-404).
- I also miss a clear connection between the parameters studied. Each paragraph is as if separate. It would be nice to present the relationships between them.
Response 10: We have carefully revised the revised manuscript. In addition, Figure 10 clearly shows the connections between the studies. Thank you for your review.
Reviewer 4 Report
Dear Authors,
First I would like to congratulate you for your work.
The manuscript is well written and the topic is very interesting.
Introduction.
The intro is good. The literature presented is adequate and the hypothesis is plausible. I only make a few suggestions in order to improve the reader's understanding.
1) The phrase “It's mainly diagnosed by laparoscopic surgery” could be rewritten taking into account that most societies recommend presumptive diagnosis, and that many lesions are more easily identified by ultrasound than by laparoscopy.
2) “Endometrium contains stem/progenitor cells, which may explain why retrograde menstruation occurs in most women…”: the presence of stem cells does not justify retrograde menstruation. Please review the sentence.
3) “abnormal endometrial stem/progenitor cells enter the pelvic cavity to cause endometriosis [4]. Endometrial stem/progenitor cells retrograde to the pelvic cavity is a prerequisite for endometriosis [5]. At the same time, the appropriate immune microenvironment in the abdominal cavity promotes the formation of endometriosis”: these sentences give the impression that these matters are defined, but in reality, they are inferences that may or may not be correct. I recommend adding some conditional. This will bring more realistic information to the reader.
Methods
The authors state that all women had a regular menstrual cycle. This is not expected for women with endometriosis. Irregularity is frequent and many use hormonal contraceptives which can induce amenorrhea or some irregular bleeding pattern.
Please advise why women were not using hormonal contraceptives. Were all infertile? None had pain?
Please inform at which stage of the menstrual cycle the endometrial sample was obtained.
Please state the criteria used to consider differently expressed miRNAs. What p-value is considered? Which logFC? Values ​​have been adjusted for multiple tests (e.g. FDR?)
Was an enrichment analysis performed to ascertain the plausibility of the findings?
Results
In my opinion, the figures are of excellent quality and are self-explanatory.
Discussion
The discussion is relevant. The limitations are briefly addressed, but they are sufficient as long as some questions are answered in the methods section.
Author Response
Introduction.
The intro is good. The literature presented is adequate and the hypothesis is plausible. I only make a few suggestions in order to improve the reader's understanding.
1) The phrase “It's mainly diagnosed by laparoscopic surgery” could be rewritten taking into account that most societies recommend presumptive diagnosis, and that many lesions are more easily identified by ultrasound than by laparoscopy.
Response 1: Thanks for your thoughtful suggestion. We have rewritten this sentence (Page 1, Line 31-32). It can be diagnosed by clinical diagnosis, imaging techniques and surgical diagnosis.
2) “Endometrium contains stem/progenitor cells, which may explain why retrograde menstruation occurs in most women…”: the presence of stem cells does not justify retrograde menstruation. Please review the sentence.
Response 2: These sentences have been changed in the revised manuscript (Page 1, Line 37-39). Endometrium contains stem/progenitor cells. Studies have found that endometrial stem cells from endometriosis patients show stronger abilities of proliferation, migration, and angiogenesis.
3) “abnormal endometrial stem/progenitor cells enter the pelvic cavity to cause endometriosis [4]. Endometrial stem/progenitor cells retrograde to the pelvic cavity is a prerequisite for endometriosis [5]. At the same time, the appropriate immune microenvironment in the abdominal cavity promotes the formation of endometriosis”: these sentences give the impression that these matters are defined, but in reality, they are inferences that may or may not be correct. I recommend adding some conditional. This will bring more realistic information to the reader.
Response 3: These sentences have been changed in the revised manuscript (Page 1, Line 39-41). Endometrial stem cells may reflux into the pelvic cavity via retrograde menstruation and interact with the peritoneal immune microenvironment, leading to the occurrence of endometriosis.
Methods
The authors state that all women had a regular menstrual cycle. This is not expected for women with endometriosis. Irregularity is frequent and many use hormonal contraceptives which can induce amenorrhea or some irregular bleeding pattern.
Please advise why women were not using hormonal contraceptives. Were all infertile? None had pain?
Response: The EMs group included the patients who underwent laparoscopic surgery for ovarian endometriosis. We don't mean that patients with endometriosis don't use hormonal contraceptives. Because endometriosis is an estrogen-dependent disease and we needed to collect endometrial tissues for culture. To eliminate the interference of abnormal menstruation and hormonal contraceptives, patients with regular menstrual cycles and who had not taken any hormone drugs within 6 months were included, while those patients who had irregular menstrual cycles or used hormonal contraceptives within 6 months were excluded in our study.
Please inform at which stage of the menstrual cycle the endometrial sample was obtained.
Response: Patients at the proliferative phase and the secretory phase were included. This couldn't affect the results obtained. We compared the protein and RNA concentrations of exosomes between patients at the proliferative phase and the secretory phase. There was no statistical difference.
Please state the criteria used to consider differently expressed miRNAs. What p-value is considered? Which logFC? Values have been adjusted for multiple tests (e.g. FDR?)
Was an enrichment analysis performed to ascertain the plausibility of the findings?
Response: We have added the criteria used to consider differently expressed miRNAs (Page 3, Lines 140-141). Differential expression miRNAs were screened with Fold change (FC) ≥1.50 and P<0.05 as criteria. We have adjusted for multiple tests and performed an enrichment analysis. In addition, the differentially expressed miRNA we studied has been verified by RT-qPCR to confirm the accuracy of the results (Figure 7C, 8C).
Results
In my opinion, the figures are of excellent quality and are self-explanatory.
Response:Thank you very much for your positive comment on our manuscript.
Discussion
The discussion is relevant. The limitations are briefly addressed, but they are sufficient as long as some questions are answered in the methods section.
Response:Thank you very much for your positive comment on our manuscript.
Round 2
Reviewer 2 Report
The manuscript has been improved and can be accepted in the present form.
Author Response
Thank you very much for your positive comment on our manuscript.
Reviewer 3 Report
The purpose of Jiang et al. was to investigate the role of exosomes in pathogenesis of endometriosis. This is very interesting paper, however I have only few questions:
1. Still, the introduction is too extensive. Especially the first paragraphs, where the authors present general information about endometriosis and disease pathogenesis. Why do the Authors write about stem/progenitor stem cells so extensively? Please show connection with the assumptions of the experiment. Additionally, there is not only one theory of pathogenesis of endometriosis.
2. The general, introduction or discussion as well need editorial improvement. Some information is repeated several times.
3. Perhaps it is worth adding information that the phase of the menstrual cycle did not affect the determined parameters.
4. Is it possible that uterine aspirate fluid reflects the state of eutopic endometrium? It has been reported that M1 macrophages are enriched in the eutopic endometrium, whereas the macrophages of the ectopic endometrium are polarized toward an M2-type” Li MZ, Wu YH, Ali M, Wu XQ, Nie MF. Endometrial stromal cells treated by tumor necrosis factor-α stimulate macrophages polarized toward M2 via interleukin-6 and monocyte chemoattractant protein-1. J Obstet Gynaecol Res. 2020;46:293-301. 10.1111/jog.14135.
5. Is there a difference between the studied parameters in the uterine aspirate fluid and the peritoneal fluid or blood of women with endometriosis?
6. What is the evidence for this transfer of exosomes from the uterine cavity to the peritoneal cavity? I understand that only retrograde menstrual blood flow?
7. ,
Author Response
Please see the attachment. Thank you very much for your comments and suggestions.
